# Evaluating Factors That Influence Influenza Vaccination Uptake among Pregnant People in a Medically Underserved Area in Washington State

**DOI:** 10.3390/vaccines12070768

**Published:** 2024-07-13

**Authors:** Kimberly McKeirnan, Damianne Brand, Megan Giruzzi, Kavya Vaitla, Nick Giruzzi, Rose Krebill-Prather, Juliet Dang

**Affiliations:** 1Pharmacotherapy Department, Spokane Campus, Washington State University, Spokane, WA 99202, USA; kavya.vaitla@wsu.edu; 2Pharmacotherapy Department, Yakima Campus, Washington State University, Yakima, WA 98901, USA; dbrand@wsu.edu (D.B.); megan.giruzzi@wsu.edu (M.G.); nicholas.giruzzi@wsu.edu (N.G.); 3Social and Economic Sciences Research Center, Pullman Campus, Washington State University, Pullman, WA 99164, USA; krebill@wsu.edu; 4CSL Seqirus, Summit, NJ 07901, USA; juliet.dang@seqirus.com

**Keywords:** influenza vaccination, pregnancy, vaccine hesitancy, perceptions of influenza vaccine efficacy, concerns about influenza vaccine safety, underserved population, Hispanic community

## Abstract

Introduction: Despite substantial evidence demonstrating the effectiveness of influenza vaccines, only 38.6% of the adult United States population received an influenza vaccine during the 2023–2024 flu season. Vaccination rates are typically lower among U.S. minority groups, and in 2022, pregnant persons from U.S. minority racial and ethnic groups showed a decrease in influenza vaccine coverage. Methods: A survey was conducted with residents of Yakima County, Washington, which is home to one of the state’s largest percentages of people who identify as Hispanic or Latino/a. The objective was to evaluate the uptake of influenza vaccine among pregnant persons. Surveys were sent to a random sample of 3000 residential mailing addresses. Of the 500 respondents, 244 (52.1%) reported that they had been pregnant, with those identifying as Hispanic or Latino/a constituting 23.8% of this total. Only 62 (26.2%) reported being immunized against influenza during pregnancy. Respondents who were immunized against influenza chose to be vaccinated to protect themselves from the flu (85.5%, n = 53); because a healthcare provider recommended getting vaccinated (85.5%, n = 53); to protect the baby from the flu (82.3%, n = 51); because it was available for free or low cost (62.9%, n = 39); and because vaccination was convenient (54.8%, n = 34). Qualitative evaluation identified that participants who were not vaccinated against influenza during pregnancy believed the vaccination was not needed, was not recommended by a healthcare provider, was difficult to access, they were against vaccination in general, or they were concerned about the safety and ingredients of the vaccine. Conclusion: Barriers to vaccination identified in this study included vaccine distrust, lack of awareness, and concerns about vaccine efficacy and safety. Healthcare providers can help address these concerns by providing education and recommendations about the importance of influenza vaccination during pregnancy.

## 1. Introduction

The U.S. Centers for Disease Control and Prevention estimate that from 1 October 2023 until 6 April 2024, there were 33 to 60 million cases of influenza illnesses, resulting in 15 to 28 million influenza-related medical visits, 360,000 to 750,000 hospitalizations, and between 23,000 and 66,000 flu-related deaths [1]. Despite substantial evidence demonstrating the effectiveness of influenza vaccines [2], only 38.6% of the United States population aged 18 years and older received the vaccine during the 2023–2024 influenza, or “flu” season [3]. Adults 18 and older experienced an initial increase in flu vaccination coverage in the season immediately following the start of the COVID-19 pandemic, and coverage has steadily declined since the 2019–2020 season and now appears to be similar to the pre-COVID numbers [4].

The flu can be deadly, particularly among high-risk patient groups, potentiating the severity of illness and worsening concurrent diseases [5]. Pregnant women are at a higher risk of severe illness associated with the flu [6]. Having the influenza infection during pregnancy was not only associated with a reduction in the average birthweight of full-term newborns but also an increased risk of late pregnancy loss (defined as pregnancy loss after 13 weeks gestation) [6,7]. The highest incidence of influenza infection was during the first trimester, and the risk of infection increased with each month that women were pregnant during the influenza season. The highest rate of hospitalization and risk of severe complications was seen during the second and third trimesters [6]. This would suggest pregnant women to be vaccinated either before the influenza season begins or as early in pregnancy to protect both mothers and their babies throughout pregnancy [6]. Pregnant people with respiratory illness symptoms experiencing fever were found to have an increased risk of preterm birth and birth defects thought to be due to the infectious process propagating a hyperactive immune and inflammatory response, [6,8]. This is particularly troubling, as nearly 50% of reported influenza-associated hospitalizations among those of childbearing-age occurred in women who were pregnant [9]. Receiving an influenza vaccine has demonstrated a reduced risk of influenza-related complications and improved outcomes among those who are pregnant, considering pregnancy and postpartum [8]. 

Vaccination rates in the U.S. are typically even lower among minority groups, such as people who identify as Hispanic or Latino/a [10]. Vaccine hesitancy and mistrust among minority groups is a complex and multifaceted issue in the U.S. A history of medical mistreatment, healthcare inequities, and unethical medical research has led to institutional mistrust and skepticism regarding even ethnically and racially targeted vaccination campaigns [11]. Concerns about vaccine safety and side effects, skepticism about vaccine effectiveness, and an underrepresentation of minority populations in vaccine trials were elements found to contribute to COVID-19 vaccine hesitancy among minority groups, according to a 2023 systematic review [12].

Although influenza vaccination rates among minority groups in the U.S. have remained steady over the last 10 years, there is a subpopulation within this statistic that are on a decline in influenza vaccine uptake. In 2022, pregnant persons from minority racial and ethnic groups in the U.S. showed a continued pattern of stagnant or, in some cases, a decrease in influenza vaccine coverage, with non-Hispanic Black pregnant women 15% below the prior year’s vaccination rates compared to their White and Hispanic counterparts [13]. The lack of influenza vaccine coverage in this demographic is further amplified when accounting for racial and ethnic group differences. In 2022, Black and Hispanic/Latina pregnant persons were 1.2- to 1.8-times more likely to undergo influenza-related hospitalization than their White counterparts [14].

Yakima County, located in central Washington State, has a population of approximately 257,000 residents. This county is home to the state’s third largest percentage of people who identify as Hispanic or Latino/a at 52%, which is significantly higher than the entirety of Washington State at 14% and the United States’ national average of 19% [15]. The term Hispanic is used by the U.S. Census Bureau as a pan-ethnic term representing people who speak Spanish and/or are from Spanish-speaking countries in Latin America, and Latino/a is a term for people who are from Latin American countries regardless of spoken language [16]. Yakima County also has Washington State’s highest percentage of people who are uninsured at 15% [15]. Much of Yakima County is considered rural as well as medically underserved [17], which is a designation assigned to counties with a shortage of healthcare services [18]. This designation underlines many of the barriers associated with lower health equity, such as lack of adequate access to health providers and in turn pre-natal care, language and cultural opportunities for education, and the seemingly universal disparity in vaccine uptake reflected throughout the nation. According to the CDC, during the 2023–2024 flu season, 24.6% of people in Yakima County were vaccinated against influenza, compared to 30.9% in Washington State and 38.6% in the U.S. [19]. A study completed in 2021 measuring vaccination uptake by women in 19 states including Washington State found that the prevalence of influenza vaccination was lowest among women identifying as Hispanic and rural and those with no health insurance. Influenza vaccination rates were also markedly lower for women who received no recommendation by a healthcare provider [13].

In 2021, there were 3370 individuals who gave birth, with approximately 53% identifying as Hispanic [20]. When comparing to other counties in the state, Yakima County has the largest average family size in household at 3.6, along with the highest birth rate in the state at 6.7% for women aged 15–50 [21]. About 12% of births in Yakima County are by 15–19 year olds, which is also among the highest percentage in the state [21]. The objective of this research was to evaluate the uptake of influenza vaccines among pregnant persons living in Yakima County. 

## 2. Materials and Methods

### 2.1. Study and Research Tool Design

A survey was developed by faculty from the College of Pharmacy and Pharmaceutical Sciences at Washington State University (WSU) to evaluate the uptake of influenza vaccine among pregnant persons. Survey questions were developed after conducting a literature search about influenza vaccine hesitancy and barriers [22,23,24,25,26,27]. Faculty collaborated with the Social Economic Sciences Research Center (SESRC) team from WSU, who provided expertise regarding question wording and survey organization. After the questionnaire was developed, it was reviewed by two vaccination industry experts who provided feedback and suggestions. After expert review, paper and electronic versions of the survey were piloted and finalized by the research team. The survey included 64 questions, of which 25 are included in this manuscript, as shown in Table 1. The survey questions have been re-numbered from the original version to reduce confusion. The results for the remaining survey questions will be reported in future publications. 

### 2.2. Participants and Data Collection

The mixed-mode web and paper survey was fielded by the SESRC team. SESRC purchased a simple random sample of 3000 resident mailing addresses within Yakima County, Washington, from the United States Postal Service Computerized Delivery Sequence File. Names of residents were not included in the purchased sample, only mailing addresses. The survey and contact materials were provided in English only. The team had discussed offering material in Spanish also but ultimately chose not to due to cost and the SESRC’s recent experience showing that very few respondents were likely to use the Spanish translation to complete the survey. Potential respondents were contacted up to five times to ask for their participation: a notification letter containing a one-dollar pre-incentive and survey link, a paper survey questionnaire packet one week later, a postcard reminder one week after the paper questionnaire, and a replacement questionnaire three weeks later. Each mailed item also included a web link for the participant to complete the survey online if they preferred. Each contact included a statement informing participants that the survey was voluntary and confidential, and results would be combined so that no one individual can ever be identified. Consecutive sampling was utilized until 500 surveys were completed, which represents an 18.3% response rate (with 152 of the 3000 addresses determined to be ineligible) and a +/−4.4% sample error. Demographic-based questions concerning race and ethnicity used standard nomenclature reflected in standardized larger yearly consensus data used as comparison in this study. An example would be the use of “Hispanic or Latino/Latina” to be inclusive of populations that have historically responded to either Latino/Latina in relation to Latin American geography origin or “Hispanic” to reflect a connection to the Spanish language. The overall survey communications and protocols followed the survey best practices guidelines from the American Association of Public Opinion Research [28]. 

### 2.3. Data Analysis

The survey results were aggregated by the SESRC team and exported into SPSS and Strata for analysis. The survey dataset was weighted by age using US Census data in order to improve the representation of respondents to the characteristics of the general population in Yakima County. A chi-square test of independence was performed to evaluate the relationship between influenza vaccination status during pregnancy and other survey items. For items where >20% of the expected cell counts were less than five, a Fisher’s Exact Test was used instead of the chi-square test. Categorical variables yielded by the demographic survey data were analyzed using descriptive statistics (frequencies).

Participants who responded that they had been pregnant but were not vaccinated against influenza were asked the open-ended question, “why did you not get the flu vaccine during a pregnancy or when trying to become pregnant?”. Responses were analyzed using qualitative inductive coding procedures [29,30]. Inducting coding involves developing codenames directly from responses themselves rather than using a predetermined codebook or framework [30]. Responses were organized into smaller samples and given code names that applied to the responses in the sample. Code names were then reviewed and organized into broader themes. 

### 2.4. Ethical Considerations

The Washington State University Human Research Protection Program found this research to satisfy the criteria for Exempt (IRB#19877-001).

## 3. Results

### 3.1. Demographics

Of the 500 survey respondents, 243 (48.6%) reported that they had been pregnant (Question 3), and their responses to the demographic questions (Questions 4–12) are shown in Table 2. 

### 3.2. Evaluating the Relationship between Vaccination and Pregnancy

Of the 243 respondents who reported they had been pregnant, only 79 (33.4%) reported being immunized against influenza during pregnancy (Question 13). The respondents who reported being immunized against influenza while pregnant were asked to select reasons they chose to be vaccinated (Question 14A). The results showed that respondents believed it was important or very important to protect themselves from the flu (88.3%, n = 213); protect family and friends from the flu (86.3%, n = 209); protect their community (72.2%, n = 175); because it was available for free or low cost (58.7%, n = 143) and convenient (58.2%, n = 142); because a healthcare provider recommended it (48%, n = 117); because family/friends recommended it (28.8%, n = 70); an employer encouraged vaccination (28.4%, n = 69); and because an employer mandated getting vaccinated (14.3%, n = 35). A few respondents wrote in other reasons for getting vaccinated against influenza, which included wanting to stay healthy, having a lower functioning immune system, feeling healthier when they received the vaccine, and because they had just always gotten it. Evaluation of the relationship between influenza vaccination status during pregnancy and reasons for choosing to get the flu vaccine within the past five years showed that none of the reasons for choosing to get the influenza vaccine had a statistically significantly relationship with influenza vaccination status during pregnancy.

Participants who responded that they had not been vaccinated against influenza during pregnancy were asked to indicate why they did not get vaccinated against influenza during pregnancy or when trying to become pregnant (Question 14B). Respondents reported that they do not believe the vaccine works (49%, n = 119), were worried about the side effects (41.5%, n = 100), it was inconvenient to get vaccinated (20.6%, n = 50), they were allergic to the vaccine (17.0%, n = 41), and for religious reasons (13.8%, n = 34) as important or very important reasons they were not vaccinated.

Qualitative analysis was conducted on open-ended responses to the question regarding why participants had not been vaccinated against influenza during pregnancy. The resulting themes included feeling that the vaccine was not needed, a lack of awareness, lack of access, being strongly against the vaccination in general, and concerns about vaccine safety and side effects. Themes and illustrative quotes are displayed in Table 3.

When asked how often within the past five years they were vaccinated against influenza (Question 15), 38.9% (n = 93) of people who had been pregnant indicated they chose to be vaccinated every year, as shown in Table 4. In contrast, 31.6% (n = 75) indicated that they had not received a flu vaccination at all within the past five years. 

Survey respondents were asked a set of questions related to how often they choose to receive routine vaccines (Questions 16–20). The majority (81%, n = 192) indicated being up to date on their routine vaccines. Of those who were up to date, 89% (n = 181) also indicated they planned to continue to receive routine vaccinations (Question 17). Those who were not up to date on routine vaccines were asked their main reason for not being routinely vaccinated (Question 18). Reasons described by respondents included concerns that vaccines do not work or are not necessary, having the intention to get vaccinated but limited access to healthcare, a lack of awareness of the need for vaccinations, a distrust of vaccines or the vaccination industry, and concerns about having a negative reaction to the vaccine. An additional 2.3% (n = 5) of respondents indicated they have an egg allergy (Question 19). The 10.6% (n = 21) of respondents who reported being up to date on vaccinations but did not plan to continue receiving vaccinations were asked to explain why not (Question 20). Reasons given included personal preference, lack of trust in vaccines, the belief that vaccines are not needed, and already being up to date with vaccines for their age.

A chi-square test of independence was performed to evaluate the significance of the relationship between flu vaccination status during pregnancy and three other survey items at *p*-value < 0.05: are you up to date on all your routine vaccinations (Question 16), will you continue to be vaccinated with routine vaccines (Question 17), and how often in the past five years have you been vaccinated against the flu (Question 15)? Due to the small number of cases that have an egg allergy (Question 20), a chi-square analysis was not evaluated for this variable. There was a significant relationship between receiving an influenza vaccination during pregnancy and being currently up to date on routine vaccines (χ^2^ df = 1, N = 227, χ^2^ = 5.008, *p* = 0.025). Those who reported being up to date on routine vaccines were more likely to have been vaccinated against influenza while pregnant than those who were not (88.5% vs. 76.5%), as shown in Table 5. There was also a significant relationship between influenza vaccination status during pregnancy and planning to continue to receive routine vaccinations (χ^2^ df = 1, N = 193, χ^2^ = 5.840, *p* = 0.016). Those who planned to stay up to date with vaccines were more likely to have been vaccinated against influenza during pregnancy than those without plans to stay current with vaccines (95.8% versus 85.2%).

Finally, there was also a significant relationship between receiving an influenza vaccine during pregnancy and the frequency of getting the flu vaccine (χ^2^ df = 3, N = 228, χ^2^ = 34.132, *p* < 0.001). Those who were vaccinated against influenza more often in the past five years were also more likely to have been vaccinated against influenza during pregnancy than those who were not (43.6% versus 34.0%). Similarly, those who had not received an influenza vaccine within the past five years were less likely to have been vaccinated against influenza during pregnancy, (16.7% versus 40%), as shown in Table 6.

### 3.3. Evaluating the Relationship between Pregnancy, Vaccination, and Trust in Medical Care

Survey respondents were asked to indicate the level of trust they had in the advice they received from various types of healthcare providers (Question 21). The majority of the respondents who had been pregnant indicated they mostly/strongly trusted the advice of their primary care provider (86%, n = 207), specialty care provider (85.2%, n = 187), pharmacist (77.8%, n = 178), and nurse (76.3%, n = 173). Responses are shown in Table 7. However, less than half of respondents (49.3%, n = 100) indicated that their healthcare provider recommended getting the flu vaccine during pregnancy (Question 22).

Survey respondents were also asked a set of questions about vaccination information and knowledge (Questions 23–25). The majority indicated that they are mostly-to-strongly confident they have received good information about flu vaccines (77.7%, n = 187), confident in their knowledge about how the flu vaccine works (72.5%, n = 175), and confident that flu vaccines are safe (70.5%, n = 171). Responses are shown in Table 8.

A chi-square test of independence was performed to evaluate the relationship between influenza vaccination status during pregnancy and trust in, confidence about, and information about the flu vaccine. The results showed a statistically significant relationship between influenza vaccination status during pregnancy and confidence that the flu vaccine is safe (χ^2^ df = 3, N = 231, χ^2^ = 12.055, *p* = 0.007). Those who responded as mostly and very confident that the flu vaccine is safe were more likely to have been vaccinated against influenza during pregnancy than those with lower levels of confidence. Respondents whose healthcare provider recommended being vaccinated were also significantly more likely to be vaccinated against influenza during pregnancy (χ^2^ df = 1, N = 200, χ^2^ = 152.158, *p* = 0.000).

## 4. Discussion

This study found that during their pregnancies, respondents were not likely to have been vaccinated against influenza. It was found that the respondents who chose to vaccinate did it to protect both themselves and their baby from the flu. For respondents who chose not to vaccinate, a number of factors played a role, including vaccine hesitancy, availability, awareness, and concerns regarding efficacy and safety. Specifically, 49% of survey respondents believed that the vaccine does not work and 41.5% were worried about the side effects, showing that targeted efforts by healthcare providers to educate patients about vaccine safety and effectiveness could be beneficial. Respondents whose healthcare provider recommended being vaccinated were significantly more likely to be vaccinated against influenza during pregnancy, further indicating the importance of provider recommendations.

Influenza vaccination rates in pregnant women were lower than expected, at 33.4%. This rate was lower than results from a study that looked at the Trivalent Influenza Vaccine for Preventing Influenza Virus Illness Among Pregnant Women during the 2010–2011 and 2011–2012 influenza seasons, which found that the 42–63% of participants received the influenza vaccine [31]. This could be for a number of reasons. First, the average age of females who were surveyed who answered being pregnant were somewhere around 45–65 years old. Secondly, even though the influenza vaccine was first recommended by the United States public health authorities in pregnancy 1960, it was not endorsed by the CDC until 1997 [32]. The majority of the respondents may not have received information about influenza vaccine during pregnancy. This matches the responses received in Table 3, as survey respondents claimed that they had never heard of the flu vaccine being available or given during pregnancies. Additionally, the timing of their pregnancy could have affected whether they were recommended the influenza vaccine or not. The American College of Obstetricians and Gynecologists (ACOG) recommends a pregnant women receive the flu vaccine as early in the season as possible, typically October through May. However, if available in August or September, a pregnant person should consider getting the vaccine, regardless of trimester [33]. This recommendation has also seemed to change over time, where focus was on the third trimester instead of any time, potentially affecting recommendation. Some respondents did mention that their doctor did not offer or recommend the vaccine during their pregnancy. Healthcare providers may be able to aid in addressing this issue by offering recommendations that are specific to each patient about the timing of the influenza vaccine in relationship to their pregnancy and the influenza season. 

Another common theme identified was that participants did not feel the vaccine was needed. Perhaps this is secondary to lack of education. Multiple clinical trials have demonstrated efficacy in pregnant women. One study that evaluated the effectiveness of the influenza vaccine at preventing influenza-associated hospitalizations during pregnancy found that the number of unvaccinated to vaccinated patients was statistically significant (*p*-value < 0.0001), with the percentage of unvaccinated pregnant women ranging from 78 to 87%. When then assessing the effectiveness of the influenza vaccine, it was found to be 48% effective (95% confidence interval [CI]: 28%–63%) in the unadjusted group vs. 40% (95% CI: 12%–59%) in the adjusted group, which took into account site, season, timing, and presence of any high-risk medical conditions [34]. By providing evidence about the risks of contracting influenzas during pregnancy to their patients, healthcare providers may be able to address the patient belief that vaccination against influenza is unnecessary. The results also identified a significant relationship between being vaccinated against influenza during pregnancy and being up to date on all routine vaccinations, planning to continue to be vaccinated with routine vaccines. Further research is needed to identify whether encouraging a patient to be vaccinated against influenza could have a positive impact on their decision to receive other routine immunizations.

As with any survey-based research, this study has limitations. Since this survey had a limited number of respondents who all live in one geographic area, opinions, and therefore results, may differ in other communities. This survey also used a simple random sample of residential addresses in Yakima County. Theoretically, the demographic composition of the sample is representative of the population, but realistically, respondent demographics may not be an identical representation of the whole population. Additional limitations to conducting vaccination-related surveys may include survey fatigue (leading to skipped survey questions), fluctuations in vaccine availability, changes in political climate, and the influence of vaccine coverage on social media. This survey was designed by the research team after reviewing the literature on this topic, but the survey tool was not validated. Additionally, this survey was not conducted solely for the purpose of pregnancy, but instead is a portion of a larger influenza-related survey. A larger and more targeted survey may yield different results. Although the survey included qualitative components, it was generally designed to be quantitative, so quantitative research using methods such as key informant interviews or focus groups could provide more in-depth results.

Future research regarding influenza vaccination among pregnant people is needed. The results from this survey have been incorporated into academic detailing presentations which are being disseminated to local healthcare providers in Yakima County. Additionally, patient education material showing the survey results and emphasizing the importance of influenza vaccination during pregnancy is being created and distributed to local hospitals, resident homes, and pharmacies.

## 5. Conclusions

Given the benefits of influenza vaccination in pregnant persons and the disparity in their vaccination rates, particularly in minority racial and ethnic groups, there is a need to address barriers in this population. In pregnant persons who historically have not been up-to-date on vaccinations and those who no longer plan to be, the key barriers appear to be vaccine distrust, necessity, awareness, and worry concerning efficacy and safety. Healthcare providers can help address these concerns by providing education and recommendations about the importance of receiving an influenza vaccination during pregnancy and risks of influenza to the pregnant person and unborn child. Healthcare providers can also help pregnant patients by including a specific recommendation for when to be vaccinated based on their pregnancy and the timing of the influenza season. Future plans to bridge this gap should emphasize protective features of influenza vaccination for pregnant persons and their baby through healthcare provider recommendations.

## Figures and Tables

**Table 1 vaccines-12-00768-t001:** Survey questions.

Survey Question Number	Survey Question	Topic Domain
1	Have you lived in Yakima County for the past 12 months or longer?	Inclusion criteria
2	Are you age 18 or older?	Inclusion criteria
3	Have you ever been pregnant?	Inclusion criteria
4	Please indicate your ethnicity: Latino/Latina//Hispanic; Non-Hispanic.	Demographics
5	Please indicate your race: white or Caucasian; Black or African American; Asian; Native Hawaiian or Other Pacific Islander; American Indian/Alaska Native/Tribal; Other, please specify.	Demographics
6	What is the highest level of education you have completed: less than high school; high school or GED; some college/AA degree/technical certificate; bachelors degree; advanced degree (PhD, Masters, MD, etc.); other, please specify.	Demographics
7	Which of the following categories best describes your current employment status?	Demographics
8	Which of the following categories best describes who you work for?	Demographics
9	Which of the following is your age group?	Demographics
10	Have you ever served in any branch of the United States military?	Demographics
11	Which one of the following best describes your current marital status?	Demographics
12	Which gender do you identify as?	Demographics
13	Did you get the flu vaccine during your pregnancy or when you were trying to become pregnant?	Pregnancy and vaccination
14A	If you answered “yes” to the previous question, how important were each of the following reasons for you to get vaccinated against the flu: protect yourself from the flu, protect your family/friends from the flu; protect your community from the flu; your employer encouraged you to get vaccinated; your employer mandated that you get vaccinated; your family/friend encouraged you to get vaccinated; a healthcare provider encouraged you to get vaccinated; it was convenient; it was available for free or low cost; or other (please describe).	Pregnancy and vaccination
14B	If you answered “no” to the previous question, how important were each of the following reasons for you to choose not to get vaccinated against the flu in the past 5 years: Getting the vaccine is inconvenient; I do not believe the vaccine works; I am allergic to the vaccine; Religious reasons; I was worried about the side effects; or other (please describe)	Pregnancy and vaccination
15	How often in the past five years have you been vaccinated for the flu?	Influenza vaccination status
16	Are you up to date on all your routine vaccinations? Routine vaccinations include vaccines like pneumonia, hepatitis, measles, and tetanus, but does not include COVID-19 or flu vaccine.	Routine vaccination status
17	Will you continue to be vaccinated with routine vaccines?	Routine vaccination status
18	If you answered “no” to the previous question, why not? (open ended response field)	Routine vaccination status
19	Do you have an egg allergy?	Routine vaccination status
20	If you are up to date on all your routine vaccinations but answered “no” to “Will you continued to be vaccinated with routine vaccines”, please explain why not.	Routine vaccination status
21	How much do you trust the advice of each of the following healthcare providers: primary care provider, specialty care provider, pharmacist, nurse, other (please list).	Healthcare provider trust and advice
22	Was getting the flu vaccine during pregnancy recommended by your healthcare provider?	Healthcare provider trust and advice
23	How confident are you that you have received good information about the flu vaccine?	Vaccination confidence
24	How confident are you that the flu vaccine is safe?	Vaccination confidence
25	How confident are you in your knowledge of how the flu vaccine works?	Vaccination confidence

Abbreviations used: GED: General Educational Development (U.S. high school equivalency degree); AA: Associate of Arts degree; PhD: Doctor of Philosophy degree; MD: Medical Doctor degree.

**Table 2 vaccines-12-00768-t002:** Demographics of survey respondents who reported they had been pregnant.

Demographic	Number	Percentage
Age (n = 244)		
18–24 years	2	0.6%
25–34 years	24	9.9%
35–44 years	45	18.6%
45–54 years	61	24.9%
55–64 years	64	26.1%
65–79 years	34	14.1%
80 years or older	14	5.7%
Race (n = 244, multiple response possible)		
White or Caucasian	204	83.6%
American Indian/Alaska Native/Tribal	11	4.5%
Asian	5	2.0%
Black or African American	5	2.0%
Native Hawaiian or Other Pacific Islander	2	0.8%
Other (Mexican, Hispanic)	14	5.7%
Ethnicity (n = 228)		
Latino/Latina/Hispanic	54	23.8%
Non-Hispanic	174	76.2%
Highest level of education completed (n = 239)		
Less than high school	12	5.1%
High school or GED	45	19.0%
Some college/AA degree/technical certificate	83	34.6%
Bachelor’s degree	60	25.0%
Advanced degree (Masters, PhD, MD, etc.)	37	15.3%
Other (multiple degrees)	2	1.0%
Current marital status (n = 245)		
Married	150	61.4%
Living with a partner	14	5.6%
Never married	13	5.1%
Divorced or separated	42	17.4%
Widowed	23	9.3%
Other (single, living alone, married but living separately)	3	1.1%
Employment status (n = 241)		
Employed full-time	136	56.8%
Employed part-time	19	7.9%
Not employed, looking for employment	2	0.6%
Not employed, not looking for employment	26	10.7%
Retired	58	24.0%
Employment Category for those employed (n = 155)		
Private, for-profit company or business, or for an individual	61	39.6%
Private not-for-profit, tax-exempt, or charitable organization	13	8.1%
Local government (city, county, etc.)	15	9.8%
State government	33	21.2%
Federal government	0	0.3%
Self-employed in my own NOT incorporated business, professional practice, or farm	14	8.9%
Self-employed in my own incorporated business, professional practice, or farm	8	5.2%
Working without pay in family business or farm	2	1.0%
Other (healthcare, higher education, school district)	9	5.9%
Ever served in the United States military (n = 243)		
Yes, now on active duty	0	0.0%
Yes, on active duty in the p0.0ast, but not in last 12 months	6	2.6%
No, training for Reserves or National Guard only	0	0.0%
No, never served in the military	237	97.4%

Abbreviations used: GED: General Educational Development (U.S. high school equivalency degree); AA: Associate of Arts degree; PhD: Doctor of Philosophy degree; MD: Medical Doctor degree.

**Table 3 vaccines-12-00768-t003:** Themes and illustrative quotes from participants regarding why they were not vaccinated against influenza while they were pregnant or trying to become pregnant.

Theme	Illustrative Quote
Belief that vaccination was not needed.	“I was young and never really got sick that much”. “The flu vaccine was not important to me at that time”. “I don’t need a flu vaccine, we get the flu every test whether we are vaccinated for the flu or not”. “I didn’t feel I was at risk”.
Lack of awareness or recommendation.	“Was younger and I did not know much about flu vaccine”. “It was never offered to me by my doctors”. “Never heard of any flu vaccine given at that time of pregnancies”.
Lack of access to the influenza vaccine.	“I don’t recall a flu vaccine being available when I was pregnant in the early 1980′s”. “Lived remote in Alaska”. “I don’t believe it was available”
Did not want to be vaccinated against influenza.	“I’ve never gotten a flu shot! Never wanted to”. “Felt that it would be ok if I got [the flu], and my body would fight it”. “Never ever believed in the flu vaccine”.
Concerned about safety of vaccination during pregnancy.	“I didn’t take any Rx then or want to jeopardize pregnancy viability”. “Was afraid of what it might do to my babies”. “I did not want to risk adverse side effects to the fetus”. “Too many side effects that could harm a baby”. “I wasn’t going to put foreign chemicals in my child”.

**Table 7 vaccines-12-00768-t007:** Survey respondents reported trust in healthcare providers.

How Much Do You Trust the Advice of Each of the following Healthcare Providers? (Question 21)	Not at All	Only a Little	Mostly	Strongly
Primary care provider (n = 241)	3.8%	10.3%	46.5	39.9%
Specialty care provider (cardiologist, pulmonologist, etc.) (n = 219)	8.0%	6.8%	43.4%	41.8%
Pharmacist (n = 229)	6.1%	16.0%	45.9%	31.9%
Nurse (n = 228)		23.8%	46.7%	29.5%

**Table 8 vaccines-12-00768-t008:** Survey respondent confidence in influenza vaccination information and knowledge.

Survey Question	Not at All Confident	Only a Little Confident	Mostly Confident	Very Confident
How confident are you that you have received good information about flu vaccines? (Question 23, n = 241)	8.9%	13.5%	47.7%	30.0%
How confident are you in your knowledge of how the flu vaccine works? (Question 24, n = 241)	8.9%	19.6%	48.7%	23.8%
How confident are you that flu vaccine is safe? (Question 25, n = 242)	15.4%	14.1%	40.9%	29.6%

**Table 4 vaccines-12-00768-t004:** Frequency of being vaccinated for influenza in the last five years among women who were or had previously been pregnant.

Frequency of Being Vaccinated against Influenza in the Last 5 Years Question 15 (n = 238)	Number	Percentage
Not at all	75	31.6%
1–2 times	34	14.4%
3–4 times	36	15.1%
Every year	93	38.9%

**Table 5 vaccines-12-00768-t005:** Relationship between being vaccinated during pregnancy and routine vaccination use survey question responses.

Survey Item		Received Influenza Vaccine during Pregnancy or When Trying to Become Pregnant (Question 13)				Total	
		Yes		No			
		Number	Percentage	Number	Percentage	Number	Percentage
Are you up to date on all your routine vaccinations? (Question 16) (n = 227)	Yes	69	88.5%	114	76.5%	183	80.6%
	No	9	11.5%	35	23.5%	44	19.4%
Will you continue to be vaccinated with routine vaccines? (Question 17) (n = 193)	Yes	68	95.8%	104	85.2%	172	89.1%
	No	3	4.2%	18	14.8%	21	10.9%

**Table 6 vaccines-12-00768-t006:** Relationship between being vaccinated during pregnancy and frequency of getting vaccinated against influenza in the past five years.

		Received Influenza Vaccine during Pregnancy or When Trying to Become Pregnant (Question 13)				Total	
		Yes		No			
		Number	Percentage	Number	Percentage	Number	Percentage
Frequency of getting flu vaccine within the past five years (Question 15)	Not at all	13	16.7%	60	40.0%	73	32.0%
	1–2 times	6	7.7%	28	18.7%	34	14.9%
	3–4 time	25	32.1%	11	7.3%	36	15.8%
	Every time	34	43.6%	51	34.0%	85	37.3%
Total		78	100.0%	150	100.0%	228	100.0%

## Data Availability

The datasets presented in this article are not readily available because the data are part of an ongoing study. Requests to access the datasets should be directed to the corresponding author.

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
