# Peer review of "Evaluating Factors That Influence Influenza Vaccination Uptake among Pregnant People in a Medically Underserved Area in Washington State"

_vaccines, 2024, doi:10.3390/vaccines12070768_

Round 1

Reviewer 1 Report

Comments and Suggestions for Authors

Thank you for the opportunity to review your manuscript. There are enhancements that could be made for greater clarity and explanatory value. Please see the attached file.

Author Response

Thank you for taking the time to provide these comments for us. This has been very helpful and we believe has had a positive impact on the manuscript. Attached you will find our responses to the your comments and the comments of the other reviewers.

Thanks again,

Kim

Reviewer 2 Report

Comments and Suggestions for Authors

Abstract

The abstract should be composed under these titles. Introduction, methods and conclusions

Lines 31-33. The conclusions are in line with the study findings to be revisit

Comment: Could you add vaccine hesitancy as one of the keywords?

Introduction

Lines 53-54. Could add the latest vaccination rate over the last 5 years documented among minority group

Lines 57-58………..showed a decrease in influenza vaccine coverage, with some of these groups trending 15% below the prior year’s vaccination rates compared to their white counterparts [9].

Comment: some of these groups which one? Trending 15% compare to which year/s

Lines 60-61. Black and Hispanic/Latino pregnant persons were 1.2 to 1.8 times more likely to have influenza- related hospitalization than their white counterparts [10].

Comment: Which year? Add

Methods

Comment: Add consent approaches implemented

Comment: Was the questionnaire in English or Hispanic/Spanish

Lines 107-108. Consecutive sampling was utilized until 500 surveys were completed.

Comment: add response rate

Lines 114-116  A chi-square test of independence was performed to evaluate the relationship between influenza vaccination status during pregnancy and other survey items. For items where > 20% of the expected cell counts were less than five, a Fishers Exact Test was used instead of the chi-square test

Comment: Could you add a p-value of <0.05 indicates statistical significance?????

Results

Lines 1555-156………….that none of the reasons for choosing to get the influenza vaccine had a statistically significantly relationship with influenza vaccination status during pregnancy.

Comment: Could you add the table that show it’s not statistically significant?

Comment: General one please add n=? Whenever you narrated percentage ie (49%, n=?) for all the results

Comments: add the p value results in the respected tables

Comments: All tables contain under percentage % I suggest to delete % as percentage is enough. Similarly number (N=?) you could use either number or N

Discussions

Lines 248-249. It was found that the patients who choose to vaccinate did it protect both themselves and their baby from the flu

Comment: the sentence is unclear, could you revisit it?

Comment: you need to discuss all the key study findings

Conclusions

Comment: the conclusion should contain the key study findings following the way forward. Please revisit the conclusions

Author Response

(The authors gave the same response as above.)

Reviewer 3 Report

Comments and Suggestions for Authors

The research topic is interesting and timely. The research itself was designed and described clearly and may be of interest to the readers of the Journal. At the same time, some modifications and improvements are required. Below, I list some comments that I hope the Authors may find helpful:

1. While the Introduction is well written, the Authors should also describe the influenza vaccine policy/strategy in the US, especially among minorities: how was it planned and undertaken? Is it successful? What is the vaccine rate in the country? And does it differ from other countries in the region?

 2. The Introduction lacks a more detailed explanation of vaccine hesitancy. In its current form, there is little to no information on vaccine hesitancy, public mistrust, and uncertainty in science both in the US and elsewhere, which should be discussed more deeply.

3. While the Authors rightly observe that vaccine hesitancy is of particular concern among ethnic minorities, including communities of color, I would expect at least a short description of the reason behind it. It is essential to contextualize the research for the readers unfamiliar with the topic, highlight the problem (what), and explain the reasons behind it (why). Thus, I would strongly recommend adding at least a short paragraph explaining that the lower levels of trust toward science and state-sponsored health programs among such ethnic minorities as African-Americans, Mexican-Americans, Native Americans, Hawaii and Alaskan Natives often result from their previous experiences with unethical healthcare research in ethnic populations (colonization, eugenics and medical experiments), their experiences with systemic racism and discrimination, under-representation of minorities in health research and vaccine trials or negative experiences within a culturally insensitive healthcare system.

4.  The Authors have to work on the structure of the methodological part. For better reception, it is worth dividing the content of the methodological part according to the following scheme: study design, participants and setting, research tools, data collection, ethical issues, and data analysis. Some aspects must be clarified or included in their present form, reducing the work's methodological value.

5. Some methodological issues: In what language was the survey conducted? Was it English or Spanish? How could it affect the response rate? What was the sampling method? What was the response rate?

6. There are other limitations: low number of respondents, the tool not being validated, possible selection bias, and only quantitative data; thus, this topic requires in-depth qualitative studies.

7. Finally, the paper would benefit from adding some recommendations suggesting possible guidelines that should be implemented to overcome the problem discussed in the manuscript.

To conclude, I believe that the issues raised in this manuscript are essential and timely, and the paper itself fits well with the aims of the Journal. At the same time, I believe it requires some revisions and completion before it can be published.

Author Response

(The authors gave the same response as above.)

Round 2

Reviewer 2 Report

Comments and Suggestions for Authors

NIL

Reviewer 3 Report

Comments and Suggestions for Authors

I have read all the reviews, the Authors’ responses, and the revised manuscript with interest. The Authors have clarified all issues raised in the review, and I believe this revised manuscript is now more consistent owing to their corrections and additional arguments. I appreciate this effort and have no further concerns regarding the manuscript. Overall, I believe the article is important and interesting and fits well with the aims of the Journal. For that reason, I recommend its publication.